# A Safe Home? A Qualitative Study into the Experiences of Adolescents Growing Up in the Dutch Area Impacted by Earthquakes Induced by Gas Extraction

**DOI:** 10.3390/ijerph19084716

**Published:** 2022-04-13

**Authors:** Elianne A. Zijlstra, Mijntje D. C. ten Brummelaar, Mileen S. Cuijpers, Wendy J. Post, Ingrid D. C. van Balkom, Hamed Seddighi

**Affiliations:** 1Department of Behavioural and Social Sciences, University of Groningen, 9712 TJ Groningen, The Netherlands; m.d.c.ten.brummelaar@rug.nl (M.D.C.t.B.); mileen_cuypers@hotmail.com (M.S.C.); w.j.post@rug.nl (W.J.P.); 2Jonx Department of (Youth) Mental Health and Autism, Lentis Psychiatric Institute, 9728 JR Groningen, The Netherlands; idc.vanbalkom@lentis.nl; 3Campus Fryslân, University of Groningen, 7901 LB Leeuwarden, The Netherlands; h.seddighi@rug.nl

**Keywords:** earthquakes, hazards, gas extraction, adolescents, well-being, living environment

## Abstract

For decades, the Netherlands has experienced minor earthquakes due to gas extraction. This study aims to obtain insight into the experiences of adolescents and the impact of these earthquakes on their well-being and living environment. Focus groups were held with 24 adolescents, and interviews were held with 3 adolescents (N = 27; M = 15 years). Through qualitative analysis, we identified six themes. The adolescents shared experiences of anxiety related to the earthquakes and their consequences and considered these to be a normal part of their life. Anxiety and feelings of endangerment not only related to their own experiences but were also connected to the impact of earthquakes on their social environment, such as the restoration of buildings. Several sources of support (e.g., talking, social cohesion) were mentioned to deal with the negative consequences of the earthquakes. A lack of trust in the government was an additional main theme, with adolescents mentioning several needs, potentially relevant to policymakers in the Netherlands. Growing up in the gas extraction area of Groningen had many consequences on the adolescents in the study, who felt inhibited from expressing feelings of anxiety and fear. To support their needs, interventions at the individual, family, educational, societal, and policy levels are recommended.

## 1. Introduction

Since 1986, Groningen, which is located in the north of the Netherlands, has experienced many earthquakes due to gas extraction, the majority of which are considered to be minor (earthquakes of magnitude 2.0–3.9 on the Richter scale are ‘minor’ earthquakes). Earthquakes are categorised as minor, moderate, major, or extremely major according to their magnitude on the Richter scale [1]. In 2012, an earthquake of magnitude 3.6, then the largest earthquake in this region, caused great concern. Various technical studies have been conducted on these earthquakes [2,3,4]. Even though the earthquakes in Groningen do not have a devastating impact, their impact is still significant because the earthquakes are close to the surface (3 km), and a 100-metre-thick sandstone reservoir leads to subsidence [5]. These earthquakes have damaged buildings, an enormous impact on the well-being of people in Groningen [6], and major economic consequences such as declining house prices, a negative perception of the area, and loss of productivity [6,7]. The earthquakes in Groningen can be seen as hazards. Hazards have different sources, including natural hazards (floods, earthquakes, tornadoes, extreme weather events) and human-made hazards (war, explosions, technological) [8]. The all-hazard approach emphasises that while hazards may come from a variety of sources, they often challenge the health system in a similar way [9,10,11]. Therefore, risk reduction, preparedness, response, and recovery programmes should be planned regardless of the source of hazards [9,10,11].

Children’s physiological and psychological vulnerabilities to hazards [12] can lead to sleep problems [13], externalising behavioural problems such as conduct, behaviour, and hyperactivity and internalising problems such as anxiety and depression [14]. In a study on the psychological effects of drought as a slow-onset disaster on adolescents, Carnie et al. (2011) reported that adolescents showed signs of fear of the future, worries about family, increased isolation, and fear of harm [15]. Furthermore, child abuse increases following natural and human-made hazards [16]. Studies have shown that the socioeconomic, housing, and shelter status of a family and history of violence are among the predictors of child abuse in disasters [17]. Those younger than 18 years old experience different vulnerabilities, depending on their age and developmental level [18]. Therefore, the effect of stressful events such as earthquakes on young children can be different from their effect on children of older ages, such as adolescents. However, the scientific literature is not conclusive on whether children in a specific developmental phase are particularly vulnerable to hazards [19].

Psychosocial effects are one of the consequences of earthquakes. A report by Holsappel et al. (2017) reviewed 23 available studies up to 2017 about the psychosocial impact of the earthquakes in Groningen [20,21]. Analyses showed serious health effects in adults, mainly related to experiencing damage to their homes and the complexity of the handling of this. For instance, people with considerable damage to their homes reported poorer perceived health and more psychological complaints [20,21]. Psychosocial impacts, according to this report, are loss of control, lack of trust, anger, worries about the future, feelings of injustice, risk perception, perceived safety, and social cohesion [20,21]. Another study on the psychosocial impacts of earthquakes on adults showed several themes of concern, such as ‘damage to property and concern about compensation mechanisms; decline in house prices; concern about the chance of dykes breaking; feelings of insecurity; health issues; and increased distrust and anger’ [6]. According to van der Voort and Vanclay (2015), some people in their study reported mental health problems due to these earthquakes, including stress, anxiety, insomnia, and depression [6]. In addition, people experienced a stressful process when seeking earthquake damage compensation [5]; some received compensation, while others had to wait many years, which sometimes also resulted in divisions within communities. Lack of proper communication about the connection between gas extraction and earthquakes in Groningen resulted in distrust directed at the extractive company as well as at the national government [6]. The above-mentioned research was focused on adults, and little attention has been paid in research to the psychosocial impact of the earthquakes in Groningen on children. The only research into children growing up in the earthquake area of Groningen, conducted by the Netherlands’ Ombudsman for Children, found that children had specific problems, such as nightmares and sleep problems, being scared of earthquakes, worrying a lot, or being concerned about the future [22].

Although various studies have been conducted on the effects of earthquakes on adolescents’ health, these studies have focused on large earthquakes (with higher magnitudes) that caused widespread damage to people, the economy, and society. Minor earthquakes have been neglected by researchers. The aim of this study is to investigate the experiences of adolescents who have experienced gas extraction earthquakes in the Dutch province of Groningen in relation to their well-being and living environment. We decided to choose adolescents as our target group because children and adolescents experience disasters differently [23].

## 2. Materials and Methods

### 2.1. Design

The present study is part of a broader mixed methods research project focusing on the experiences of children and adolescents growing up in the gas extraction area of Groningen [24] and is funded by the National Coordinator for Groningen. For the present study, we organised focus groups and interviews with adolescents in which they could share their experiences of growing up in this area [25,26]. The Ethics Committee for Pedagogical and Educational Sciences of the University of Groningen approved the research design.

### 2.2. Participants

The study sample consisted of 27 adolescents between the ages of 12 and 20 years, with a mean age of 15 years (see Table 1). A total of 24 adolescents participated in the focus groups, while 3 were reluctant to share their experiences in a group and were interviewed individually. All adolescents grew up in the gas extraction area of Groningen with their parents and siblings. They experienced various forms of physical damage to their homes. Eighteen adolescents mentioned three or more episodes of earthquake-related damage, four experienced damage once, two experienced none, and one adolescent did not know whether damage had occurred.

### 2.3. Procedure

The research team approached secondary schools and social welfare organisations in the region and informed them about the study. Schools, social welfare and mental health organisations informed adolescents who grew up in the earthquake area about the research using a leaflet and invited them to participate in the research.

In cooperation with interested schools and social welfare organisations, we organised five focus groups and three interviews. Prior to the focus groups and interviews, adolescents were informed of the aims of the study, the research process, and how the results would be disseminated. Adolescents signed an informed consent form, and parental consent was obtained for those younger than 16. Before starting the focus groups and interviews, trained researchers (EZ, MC) talked to adolescents about the nature of the study. Exposure to earthquakes was determined by asking how often their homes had been damaged. We made agreements about privacy between adolescents, and we assured them that they were free to end their participation in the study at any time.

The focus groups were led by two researchers following a focus group guide. In the focus groups, the two researchers took a flexible approach and followed the adolescents’ stories as much as possible. The topics of the guide were: the adolescent, family, school, neighbourhood, village, and needs of the adolescents related to growing up in the gas extraction area. Open introductory questions focused on their experiences of growing up in the gas extraction area of Groningen related to the mentioned topics. The researchers asked adolescents open-ended questions focusing on their personal experiences and invited them to share their own examples. To achieve this, a variety of methods (e.g., drawing, social circles) were used to encourage adolescents to share their experiences of well-being, living circumstances, and peer interactions. For instance, we asked adolescents to draw about growing up in the gas extraction area and to explain their drawings to each other.

With the adolescents’ permission, the focus groups and interviews were recorded. The focus groups took place at school or at the social welfare organisation and lasted approximately one to two hours. Three focus groups were held at school during school time, with adolescents receiving permission from their teachers to participate. Two focus groups were held in the evening at the social welfare organisation. The three adolescents who preferred an interview were interviewed by one researcher (MC), and the interviews lasted about 45 min. The same procedure and guide were used. Aftercare for all adolescents was provided by calling the adolescents or the contact person of the school or social welfare organisation a day after the focus group or interview.

### 2.4. Data Analysis

The audio recordings were transcribed verbatim. Two researchers (M.C. and M.B.) inductively coded three transcripts independently, and, afterwards, a concept codebook was defined [23]. In the codebook, a distinction was made between codes directly related to the earthquakes and growing up in the gas extraction area and codes not directly related to these. Using the concept codebook, the remaining transcripts were coded by one researcher (M.C.), and new codes were added for new information. During this phase, frequent contact took place between the researchers, and we reflected on the codebook and the process of coding. The themes were then defined based on a reflective discussion between three researchers (M.C., M.B., and E.Z.) about the codes directly related to growing up in the gas extraction area. Within this reflective discussion, the main experiences of adolescents were central in defining the themes. Several quotes and two drawings of adolescents are presented as an illustration of the themes.

## 3. Results

The themes found in the analysis and the mutual relationships are presented in this section. The themes were: anxiety, social environment, house restoration, support, trust in the government, and needs of children. Several quotes and two drawings (Figure 1 and Figure 2) of adolescents are presented to illustrate the themes.

### 3.1. Anxiety 

Several adolescents whose homes were damaged once or repeatedly by earthquakes mentioned feelings of anxiety related to the earthquakes. They had concerns about the safety of their home and feared the widening of cracks or were worried about the collapse of ceilings or their whole house. Some adolescents felt safe in their house because it had been reinforced using external props. Adolescents said they thought about earthquakes daily because of the visible effects on their living environment: when they saw cracks in the walls of their house, when they cycled through the village and saw the damaged church, when they laid in bed and saw the cracks in the wall of their bedroom, or when confronted with politicians or news about earthquakes in the media. Additionally, adolescents talked of sad feelings when they had to move. One adolescent said that he learned to sleep less deeply so that he could run out of the house if there was an earthquake. Those who had not experienced damage to their homes were usually less fearful.

It is notable that adolescents were afraid of the possible devastating consequences of earthquakes in their future and that of others. Adolescents shared thoughts such as: ‘what if a house collapses and people are injured or killed’, ‘what if roof tiles fall on someone’s head’, ‘what if a church collapses’, and ‘what if people suffer damage and they can’t afford to buy new things?’.

“It’s not like every time I go home I think, “oh, maybe there’s another earthquake and maybe it’s collapsing now”. In the back of your mind, you know that the chance is there and it does cause some stress. Well, you also know that nothing has been done about it yet and that it is therefore no better than it was.”(16 years)

Several adolescents described the earthquakes in Groningen as a normal part of their lives and mentioned that they were used to earthquakes and damaged houses. They were used to the cracks in houses, entire streets being renovated, and the consequences of earthquakes on the construction of houses. Adolescents described what it was like to experience earthquakes. Some found it frightening, while others were glad to experience it once so that they knew how it felt.

“I don’t suffer a lot from it. We have a lot of cracks in the bathroom, through the tiles and stuff, so yeah, you know, I’m not there that often anyway”.(14 years)

**Figure 1 ijerph-19-04716-f001:**
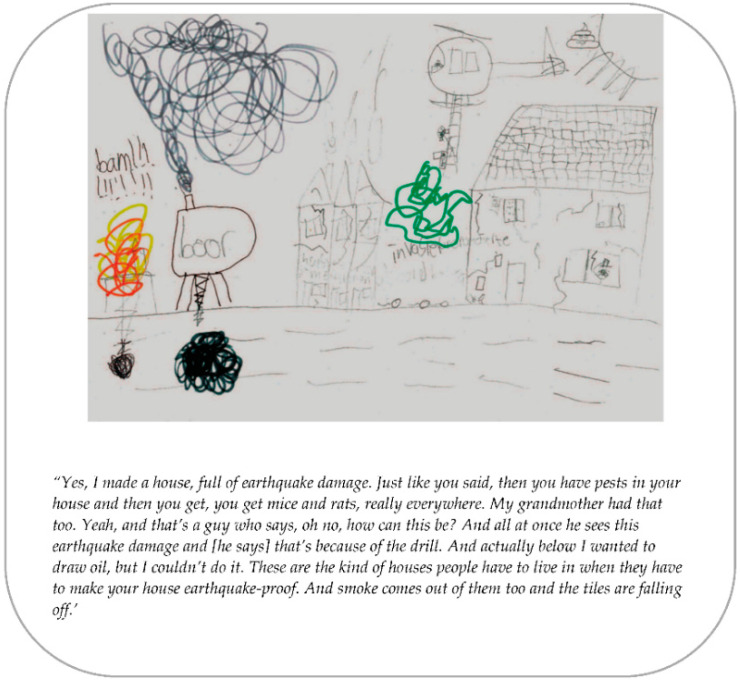
Drawing of a participant (12 years).

### 3.2. Social Environment

Adolescents mentioned the impact of the earthquakes on their social environment as creating feelings of emotional insecurity and anxiety. Adolescents indicated their parents were distressed by the earthquakes. Stressful circumstances at home, with parental attention focused on the claim procedures, resulted in parents paying less attention to their children.

“If you notice that your mother is constantly stressed when you are 12, 13, and she is very agitated, then the atmosphere at home is just bleak when you sit down to dinner at night, very agitated..…And you get that as a child and that doesn’t help at school. I think subconsciously, I think it was subconscious at the time, but looking back on it now, it did have an influence, and it was all a bit more difficult”.(19 years)

Not only at home but in the wider surroundings too, adolescents saw the effects of gas extraction and earthquakes on people. They noticed irritability, tiredness, depression, low tolerance for each other, and relationships ending in divorce. One adolescent mentioned that an adult had attempted suicide.

“When you look at what is broken, it makes me think of that old man in his house. Because the house was literally destroyed and that man was quite sad about it... Well, I think I can sympathise with people like that. What if I, if I think, well, if I were in that situation, living on my own. And if I had a beautiful little house that I was proud of, that looked nice, and it just fell down. And it had to be demolished. And you do get a very nice new house, but if that is not what you have had your whole life, what you don’t necessarily need, then it seems to me that you can be quite emotionally affected by it. You can see that in this man. When he was busy in his garden, he would say ‘Hello, good morning’. Well, now it is, I don’t really see him working in his garden because it’s not his garden anymore. It’s not his garden anymore, the one he always had”.(16 years)

### 3.3. Restoration

Adolescents talked about what happened to their homes when they were damaged by earthquakes and mentioned that repairing and strengthening houses takes a long time. Various inspectors repeatedly surveyed the damaged houses, and when they disagreed on the cause, repairs were delayed. Concerning the damage restoration, adolescents mentioned that not all cracks were repaired and that walls cracked again after new earthquakes. During repairs, adolescents were inconvenienced by the noise and the lack of privacy, in addition to parents being less attentive to their children. Several adolescents mentioned the uncertainty and annoyance of not knowing what would happen to their house and whether it would be safe to continue living there.

“Especially the uncertainty about: is there earthquake damage? What will be done about it? You know. We really had to wait two and a half months for the outcome of what would happen to our house”.(14 years)

Moving because of repairs to or demolition of their home was often mentioned by adolescents. Some had to move again because the construction of their new house or the restoration work was delayed. All of those who had to move were told they were moving to a smaller or a much smaller house, had no choice about where they moved to, and were informed shortly beforehand about when and where they were moving. Due to repairs to school buildings, children were moved to other temporary buildings. Some adolescents in secondary school expressed sadness about having to graduate from primary school in a temporary building. In their stories, adolescents expressed their emotional connection to their house, school building, and village.

“But the thing is that when you cycle past your school, you see all these fences and there is no one in the school anymore. And that feels strange because you always went to that school. It is a pity to see that there are no children playing in the playground anymore and they now have to go to another school”.(14 years)

**Figure 2 ijerph-19-04716-f002:**
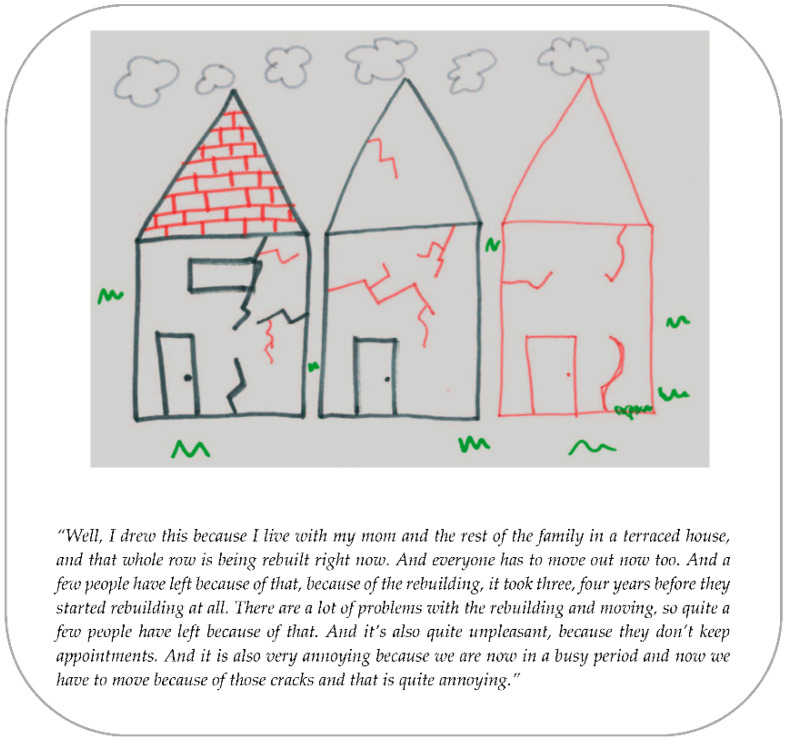
Drawing of a participant (14 years).

### 3.4. Support 

Several sources of support to deal with the negative consequences of gas extraction are mentioned by adolescents. One source of support is the acceptance of the consequences of the earthquakes as a normal part of their life. Some adolescents received support from their parents when they felt distressed, and they talked about their feelings. In other families, there were no conversations about the earthquakes and talk related to earthquakes was forbidden because of the stress it caused. Support was also found in that other people in the region struggled with the same problems. Adolescents indicated feeling supported by social cohesion when solutions were talked about in their neighbourhood or during protest demonstrations. It is notable that adolescents talk little with each other about the earthquakes and the role these play in their lives. Talking to each other about their personal experiences was not encouraged at school, and most schools focused on talks on the technical aspects of gas extraction and the earthquakes. This was evident in the focus groups and interviews where adolescents shared their knowledge about the technical aspects of the earthquakes. With a few exceptions, adolescents did not feel understood by their teachers when they talked about the personal consequences of earthquakes. Adolescents reported some teachers’ lack of understanding and over- or underappreciating the problems due to being from outside the area and not knowing what it is like to grow up with earthquakes.

The stories of adolescents reflected their need to be heard regarding their feelings and experiences concerning the earthquakes.

“Because the teacher himself does not live in that area... [and he says] ‘of course that’s annoying for those people’. Then you’re in the lesson and you think, ‘hello, I’m one of those people too.’ That’s quite, quite annoying”.(16 years)

### 3.5. Trust in the Government 

A central theme in the stories of the adolescents concerned their trust, or lack thereof, in the government’s ability to solve the problems caused by the gas extraction and the unfair treatment of people in Groningen. Adolescents suggested that the government made promises that it did not keep. This contributed to feelings of being ignored, misunderstood, betrayed, and abandoned by the national government. Adolescents did not believe that gas extraction would be stopped by the government. They also did not trust the organisation responsible for the gas extraction and were convinced that financial benefits were prioritised over the negative consequences for individuals and local communities.

Some adolescents indicated turning against the government, authorities, and politics and wanted nothing to do with them. One young person mentioned that they were hesitant about participating in this research because the university also belongs to the government. Other adolescents joined demonstrations against gas extraction.

“They are not affected by it and they don’t care about it. They just make money. And that’s how the capitalist system works. And I find that deeply sad”.(19 years)

“That something happens, that something is done about the problems. That they, that they... Promise things and then do them. Yes, when they say that they... promises are made here. No, they make promises, but they just stab you in the back”.(15 years)

Some adolescents highlighted the benefits of gas extraction (money) and that they understood the complex work of the people working for the government or for the organisation responsible for the gas extraction.

### 3.6. Needs of Children

When asked about their needs, adolescents indicated their wish for gas extraction and earthquakes to stop, as well as their worries that it might be too late to stop the earthquakes. Other adolescents gave solutions to stop the earthquakes:

“If they could invent something to stop the earthquakes, with a pump, something with the same density to pump into the ground, great!”(14 years)

Other needs included robust and safe houses and fast, clear, and fair procedures for people who have suffered damage to their homes or are at risk. Adolescents suggested that care for people must be central in the approach to solving the effects of earthquakes and that listening to citizens and their problems should be improved. Adolescents wanted to be taken seriously by the government and to be involved in solving the problems related to gas extraction. The last need mentioned by the adolescents was to present a more positive image of the province of Groningen in the media and on social media.

“People really need care and love and a listening ear... Because the confidence, because the trust, especially with the people here, especially, well, as I hear in [town], it is really, almost completely gone”.(20 years)

### 3.7. Summary: A Conceptual Model

We found that the earthquakes, especially when children were personally affected by them, caused feelings of anxiety and a lack of trust in the government and its ability to solve the ‘earthquake problem’. The needs of children result from these consequences. Witnessing the consequences of the earthquakes on the social environment and the problems with the restoration of houses and buildings reinforces these feelings for children. Support helps children to deal with these feelings. For the conceptual model see Figure 3.

## 4. Discussion

The findings show that the ongoing Groningen earthquakes induced by gas extraction have had many consequences on the lives of the adolescents studied (see Figure 3). So far, researchers have focused on the consequences of major disasters, such as severe and devastating natural earthquakes, on children’s well-being. However, the authors of this study did not find any literature on the effects of small-scale earthquakes on children’s well-being. The number of small earthquakes in the world is much higher than the number of large earthquakes, and people experience them in various locations in the world, especially in earthquake-prone areas [27]. This study found that adolescents experience anxiety and fear of the consequences of the earthquakes and that they compare these earthquakes to large earthquakes in other countries with very destructive effects. Furthermore, adolescents deal with the negative consequences of gas extraction in different ways. Although this study is about earthquakes caused by gas extraction, the difference in the source of the earthquake does not affect people’s perception. Studies have shown that stressors in everyday life, if not addressed, will gradually affect people’s mental health and have various individual, social, and economic consequences [28]. Due to the continued extraction of gas and the continuation of earthquakes, there is limited opportunity to recover from these stressors, and, therefore, the health of people, including children and adolescents, is at risk. Recourses within the family (e.g., effective parenting, continued daily routine), school (e.g., supportive schools), community (e.g., sense of belonging and pride), and policy (e.g., involving children in risk reduction of earthquakes) will protect the resilience of adolescents [23].

Participants indicated feeling inhibited by teachers and relatives from expressing their personal experiences, with more emphasis being put on technical aspects of the earthquakes. Although adolescents felt anxious, scared, and insecure about earthquakes, they did not have the opportunity to share those feelings with others. In other words, there was a lack of ‘risk communication’. The World Health Organization (WHO) states that risk communication includes ‘the range of communication capacities required through the preparedness, response, and recovery phases of a serious public health event to encourage informed decision making, positive behaviour change, and the maintenance of trust’ [29]. It is clear from this definition that stakeholders, especially the government, must put communication on their agenda at all stages of risk. In addition, various studies and standards in the field of mental health in disasters emphasise that one of the methods of psychosocial support in emergencies is communication [30].

Another issue mentioned in the findings is adolescents’ anger regarding the continuation of the earthquakes due to human activities and their destructive effects on life and the built environment. Studies have shown that prolonging the response and recovery phases has negative psychological and social consequences on the people of the region affected by earthquakes and other disasters [31]. Previous studies, as well as this study, have shown that people are upset and angry about the effects of earthquakes on their living conditions, including the partial destruction of homes.

Adolescents stated in this study that earthquakes caused their families to pay less attention to them. One of the consequences of emergencies on children’s health can be child abuse, and neglect is one of the types of child abuse that can occur [17]. Neglect refers to ‘parents, guardians or other caregivers who have access and knowledge of services but have a deficit in the provision of resources to meet the physical or mental needs of children’ [17]. Families are concerned about damage to their homes due to earthquakes, but the economic effects of falling housing prices in earthquake-prone areas are also important issues for families. All of these consequences increase the likelihood of child abuse [17]. Complex disasters such as earthquakes and the COVID-19 pandemic are more likely to cause child abuse. For instance, studies have shown that child maltreatment increased globally due to pandemic-induced stressors such as fear of infection, job and income loss, and lack of childcare [32,33,34,35,36].

Another point highlighted in this study is the impact on adolescents of earthquake damage to the home, school, and other places. Researchers have shown that places are a symbol of disaster recovery for adolescents [37]. Home and school play an important role in feeling stable, hopeful, and returning to a normal life. Place attachment has been seen among individuals of different sex, cultures, and socioeconomic status [38]. For adolescents, homes, bedrooms, and the local area are all places of interest. Adolescents feel a strong connection to their neighbourhood and home. Being forced to leave their homes due to disasters and move to a new place will have negative psychological effects on adolescents, including stress-related illness, grief, disorientation, and risky behaviours [38].

Participants indicated that they were dissatisfied with the government’s response to the Groningen earthquakes; they, therefore, did not trust the government and responsible organisations. Political trust or inclination towards government officials based on character and ability is the belief that government officials can perform their duties [38,39,40]. Trust is tied to people believing that officials are capable (people trust officials because they are competent in their job) and strong enough to do their assigned duties (people trust officials even if duties are hard for officials) [41]. Political trust is conceptualised in two ways: the first is the trust that results from beliefs and socialisation, and the second is specific trust based on experiences, such as evaluating the performance of officials during earthquakes [42]. Studies have shown that if people view disaster management as negatively in line with their expectations, it reduces their political trust [42]. Political mistrust makes it difficult for the government to advance its social programmes, especially in times of crisis. For example, studies have shown that mistrust makes it difficult for governments to enforce the stay-at-home order during the COVID-19 pandemic [40,42,43] or evacuate during floods or tornadoes [39]. If the roots of distrust are not addressed, this distrust will increase and cause serious problems for the government [40].

### 4.1. Strengths and Limitations

Because little was known about the impact of recurring earthquakes due to gas extraction on the well-being and living environment of adolescents, a qualitative design was needed to obtain a thorough understanding of the personal feelings and experiences of adolescents growing up in the earthquake area. We included adolescents with different educational levels who shared their varied experiences of growing up within or close to the earthquake area and who experienced none, one, or repeated episodes of earthquake damage to their home. With this diverse group, we have been able to provide a good overview of the potential impact of earthquakes on children, but we cannot indicate how many children in the population of children growing up in the earthquake area suffer from earthquake-related problems, as presented in this study.

In the focus groups, attention was paid to the agency of adolescents, building up trust and following their stories [44], which encouraged them to freely share their experiences. This was necessary and followed the recommendations of the Netherlands’ Ombudsman for Children (2017) study, which suggested that children growing up in the earthquake area did not readily or easily share their experiences related to earthquakes and needed trust, safety, and recognition [45].

Another strength of this study is the attention paid to the dissemination of the results. A selection of the drawings made by the adolescents was bundled in a booklet since the drawings gave an impressive illustration of the adolescents’ experiences. This booklet was widely disseminated and sent to all adolescents. A number of adolescents attended a special meeting at which the report and booklet of drawings were presented to the Dutch Minister of Economic Affairs, who is responsible for gas extraction in the Netherlands. In this meeting, adolescents had the opportunity to share their experiences directly with the responsible minister.

### 4.2. Recommendations

An important recommendation for policy and practice is to take the needs of children who grow up in the earthquake area seriously. Little attention is being paid to the impact of recurring earthquakes on the living circumstances and well-being of children in Groningen. The past has shown that the consequences of gas extraction persist over time and are not solved quickly. It is a realistic scenario that, even with reduced or discontinued gas extraction, the recovery from consequences will take a long time [46,47]. This means that children are also being exposed to the consequences of gas extraction in the long term. It is, therefore, very important that the voice and needs of children are taken into account in ‘earthquake’ policy and practice. Adolescents need the gas extraction and earthquakes to stop, and they desire safe homes, fair and fast procedures for claims, and a central position for the Groningen people in earthquake policy and practice. None of these topics seem to have been adequately addressed, while it is urgent that they are implemented [46]. From a child rights perspective, the best interests of the child should be a primary consideration in every decision that affects the lives of children, and children have the right to development and to be heard (Art. 3, 6, and 12 of the United Nations Convention on the Rights of the Child) [48]. Safeguarding these main principles in ‘earthquake’ policy and practice is needed to meet the needs of affected children.

More attention also needs to be paid to strengthening the resilience of children growing up in the Groningen earthquake area, especially those with damaged homes. This requires identifying the needs of adversely affected children and developing appropriate intervention programmes. In everyday life, it is important that children can express their feelings and questions about the gas extraction and its consequences for their individual situations. Support and reassurance will help children to gain control over their feelings and to adapt to the changing circumstances [49,50]. Support for distressed parents is also needed to enable balancing their time and attention between their children and the stressful, time-consuming procedures involved in repairing damage to their homes. This will contribute to the well-being of their children [51,52]. Accelerating the process of repairs and reconstruction of damaged buildings and ensuring transparent and timely risk communication will protect people’s resilience, including that of children [23].

Further research into the development of children who are affected by small-scale earthquakes is needed. This study has added to the knowledge of how the drawn-out small-scale earthquakes in Groningen impact the lives of children, and it contributes to the validation of the presented conceptual model. Prevalence research is needed to understand how many children are affected by the earthquakes and the factors that contribute to feelings of anxiety and trust. Longitudinal research increases our understanding of the long-term impact of earthquakes on the children’s well-being and living environment. Research into how the developmental needs of children growing up in the earthquake area may be supported is necessary to strengthen psychosocial interventions. Available international intervention programmes usually focus on the consequences of large-scale earthquakes, and more knowledge is needed to develop interventions to support children affected by minor earthquakes. For this research to be carried out in a meaningful way, cooperation with science, local welfare, and health institutions is needed as well as its prioritisation by local and national governments.

## 5. Conclusions

Growing up with earthquakes in the gas extraction area of Groningen can affect adolescents’ well-being and living environment, as the impact can lead to stress in their daily lives. Adolescents who are personally affected by earthquakes and whose homes have been damaged seem to be more affected.Despite communication being one of the methods of psychosocial support during hazards, the adolescents are not encouraged to share their personal experiences about the earthquakes at home or school or with friends.Feelings of mistrust towards local and national governments and the organisations responsible for gas extraction, due to the way in which the consequences of gas extraction are handled, should be addressed to prevent their increase.

## Figures and Tables

**Figure 3 ijerph-19-04716-f003:**
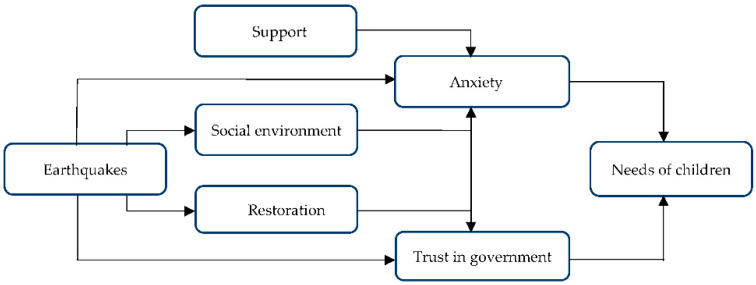
Conceptual model of themes regarding gas extraction earthquakes in Groningen from the perspective of adolescents.

**Table 1 ijerph-19-04716-t001:** Composition study sample per focus group and interviews.

	N	Age (Years)
Participants	27	12–20
Focus group 1 (school)	5	14, 14, 14, 15, 16
Focus group 2 (school)	4	16, 17, 18, 18
Focus group 3 (school)	6	12, 12, 12, 14, 14, 15
Focus group 4 (social welfare organization)	4	15, 17, 19, 20
Focus group 5 (social welfare organization)	5	12, 14, 15, 16, 19
Interviews	3	13, 14, 16

## Data Availability

The data presented in this study are available on request from the corresponding author. The data are not publicly available due to privacy of participants and traceability to participants.

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
