# Peer review of "A Safe Home? A Qualitative Study into the Experiences of Adolescents Growing Up in the Dutch Area Impacted by Earthquakes Induced by Gas Extraction"

_ijerph, 2022, doi:10.3390/ijerph19084716_

Round 1
Reviewer 1 Report
The research described is interesting and takes into account many important aspects regarding the importance of trust in institutions, risk communication, the social aspect of the "feeling" of people living in areas such as the studied one, etc ...
It's a qualitative research, but it would be very interesting if graphs were inserted to give immediate knowledge of the peculiar characteristics of the participants in the focus groups and the frequency of the answers obtained.
Information on the questions (not also the answers) collected during the focus groups would be useful to finally give a more detailed description of what the researchers did and to better understand the importance of this type of studies making them appreciate more to everyone.
The conclusions should also be more articulated and underline the strengths of this method. These are all suggestions for improving the paper.
Author Response
Reviewer 1
- The research described is interesting and takes into account many important aspects regarding the importance of trust in institutions, risk communication, the social aspect of the "feeling" of people living in areas such as the studied one, etc ...
Thank you for recognizing this important and interesting research topic. We would like to thank the reviewer for their efforts in reviewing our manuscript.
- It's a qualitative research, but it would be very interesting if graphs were inserted to give immediate knowledge of the peculiar characteristics of the participants in the focus groups and the frequency of the answers obtained.
We have added a table describing the characteristics of the participants. Since we applied a qualitative design in this study and made use of semi-structured interviews, it is not possible to add the frequency of the participants’ answers to the questions in the focus groups because the goal of the research is to obtain insight into the different types of answers and not their frequencies. We invited the children to share their experiences related to different topics (family, school, neighbourhood, village, and needs of the adolescents related to growing up in the gas extraction area). Based on the answers provided, the researchers asked more in-depth questions to explore their experiences.
- Information on the questions (not also the answers) collected during the focus groups would be useful to finally give a more detailed description of what the researchers did and to better understand the importance of this type of studies making them appreciate more to everyone.
The main purpose of this research was to obtain insight into the experiences of adolescents growing up in the gas extraction area of Groningen. In the international scientific literature, little is known about the impact of small-scale earthquakes on the living environment and well-being of adolescents. More research has been carried out on large earthquakes and their devastating impacts, but we do not know whether this knowledge also applies to adolescents affected by small-scale earthquakes. Therefore, a qualitative design was necessary with a semi-structured interview protocol and open-ended questions to explore the experiences of adolescents growing up in the earthquake area of Groningen. We have added to the manuscript, explaining that the open introductory questions were focused on the experiences of adolescents related to the topics of the interviews and that adolescents were invited to share experiences related to this.
- The conclusions should also be more articulated and underline the strengths of this method. These are all suggestions for improving the paper.
Thank you for the suggestion to emphasize our findings. We have added relevant findings as bullet points at the end of the manuscript.
- Growing up in the gas extraction area of Groningen can affect adolescents’ well-being and living environment and lead to stress in their daily lives. Adolescents who are personally affected by earthquakes and whose homes have been damaged seem to be more affected.
- Despite communication being one of the methods of psychosocial support during hazards, adolescents are not encouraged to share their personal experiences about the earthquakes at home or school or with friends.
- Feelings of mistrust towards local and national government and the organizations responsible for gas extraction due to the way in which the consequences of gas extraction are handled should be addressed, to prevent their increase.
Reviewer 2 Report
The paper “A safe home?...” is an interesting piece of work referring to an important problem. The Authors present their study well, but there are some things still lacking. Mostly due to this, at the moment the paper needs some major revisions. Nevertheless, I am looking forward for the revised version of the paper as in my opinion it should be published.
Below you may find my more or less specific comments.
Please pay attention to typing errors (for example doubled space in lines 38, 44, 134).
More information on the economic consequences of the earthquakes are needed. Just one or two sentences in the introduction would give a good view on the scale of the problem.
Line 43-44: you may also mention extreme weather events which also are rather small scale disasters and affect small communities .
Lines 49-76: there are to many irrelevant information. Please shorten this paragraph.
Lines: 112-120: You have left something from the manuscript template. Please be more attentive.
Materials and methods section.
Please elaborate more on the research procedure: you mentioned there were five focus groups – how many participants in each group, were the adolescents participating in more than one focus group, were the groups differentiated in course of participants age? Maybe a small table consisting the data would be helpful. Also please mention the issue of time – when the focus groups took place, how long etc.
Recommendations section.
You are presenting some recommendations based on your findings which I see as a very positive aspect of the manuscript. Nevertheless, I would like you to add a paragraph on local/regional policy concerning dangerous gas extraction in the area. If something is to be changed show the present state.
A conclusions section is lacking. Please prepare a short section containing major findings. You may present it just in bullets.
Author Response
Reviewer 2
- The paper “A safe home?...” is an interesting piece of work referring to an important problem. The Authors present their study well, but there are some things still lacking. Mostly due to this, at the moment the paper needs some major revisions. Nevertheless, I am looking forward for the revised version of the paper as in my opinion it should be published. Below you may find my more or less specific comments.
We would like to thank the reviewer for their compliments and efforts in reviewing our manuscript. With the help of the reflective and critical comments of the reviewers, we think we have improved the manuscript.
- Please pay attention to typing errors (for example doubled space in lines 38, 44, 134).
We have reread the manuscript carefully and corrected the errors.
- More information on the economic consequences of the earthquakes are needed. Just one or two sentences in the introduction would give a good view on the scale of the problem.
We agree with your comment and have added the following sentences about the economic consequences in the introduction: ‘These earthquakes damage buildings, have an enormous impact on people’s well-being in Groningen, and have major economic consequences such as declining house prices, a negative perception of the area, and loss of productivity.’ We reflect on this topic in the discussion by discussing it as a risk factor for child abuse.
- Line 43-44: you may also mention extreme weather events which also are rather small scale disasters and affect small communities.
We have added extreme weather events as an example of a natural hazard.
- Lines 49-76: there are too many irrelevant information. Please shorten this paragraph.
Thank you for your critical eye. We have shortened this paragraph and we think the message is clearer now.
‘Children’s physiological and psychological vulnerabilities to hazards [12] can lead to sleep problems [13], externalizing behavioural problems such as conduct behaviour and hyperactivity, and internalizing problems such as anxiety and depression [14]. In a study on the psychological effects of drought as a slow-onset disaster on adolescents, Carnie et al. (2011) reported that adolescents showed signs of fear of the future, worries about family, increased isolation, and fear of harm [15]. Furthermore, child abuse increases following natural and human-made hazards [16]. Studies have shown that the socioeconomic, housing and shelter status of a family and history of violence are among the predictors of child abuse in disasters [17]. Those younger than 18 years old experience different vulnerabilities, depending on their age and developmental level [18]. Therefore, the effect of stressful events such as earthquakes on young children can be different from their effect on children of other ages, such as adolescents. However, the scientific literature is not conclusive on whether children in a specific developmental phase are particularly vulnerable to hazards [19].’
- Lines: 112-120: You have left something from the manuscript template. Please be more attentive.
Our apologies for this omission. We have deleted this paragraph.
- Materials and methods section. Please elaborate more on the research procedure: you mentioned there were five focus groups – how many participants in each group, were the adolescents participating in more than one focus group, were the groups differentiated in course of participants age? Maybe a small table consisting the data would be helpful. Also please mention the issue of time – when the focus groups took place, how long etc.
We have added a table describing the study sample and the requested information about the procedure in the focus groups.
- Recommendations section. You are presenting some recommendations based on your findings which I see as a very positive aspect of the manuscript. Nevertheless, I would like you to add a paragraph on local/regional policy concerning dangerous gas extraction in the area. If something is to be changed show the present state.
In line with the needs of adolescents, we have added more information on what is needed in policy to meet the needs of affected children and their families.
‘Adolescents need the gas extraction and earthquakes to stop, and they desire safe homes, fair and fast procedures for claims, and a central position for the Groningen people in earthquake policy and practice. None of these topics seem to have been adequately addressed, while it is urgent that they are implemented.’
- A conclusions section is lacking.Please prepare a short section containing major findings. You may present it just in bullets.
We have added a conclusion section with the main findings presented in bullet points.
Reviewer 3 Report
I carefully read the ms by A.E. Zijlstra et al. entitled "A safe home? A qualitative study into the experiences of adolescents growing up in the Dutch area impacted by earthquakes induced by gas extraction".
The topic is quite interesting, and the survey among adolescents, although the very reduced number of participants, seems well conducted.
Notwithstanding this, I have some main concerns regarding presentation of the ms, mostly because of several sentences that are obscure to me.
Although I am not a native English speking, what I fell in most paragraphs is that many redundant expressions are used, which make reading quite difficult. I would have appreciated the ms much more if, in general, the authors hade made a better effort toward schematizationof results. I can suggest to add a table at the beginning of the Discussion section to summarise the possible implications deduced from the answares to the different questions of the survey.
In as much, the authors left in the ms some paragraphs that do not pertain to it, like what I wonder wether is a part of Journal's Istruction to authors or not, or a couple of sentences that had already been written elsewhere in the text. An other problem I encountered concerns the drawings, which are recalled along the text, but that I did not find anywhere...possibly, it was my fault.
The only attached material I found was a document in Dutch I cannot understand. If the drawings the authors make reference to are part of that document, I feel too difficult to find the right one in the present form.
Author Response
Reviewer 3
- I carefully read the ms by A.E. Zijlstra et al. entitled "A safe home? A qualitative study into the experiences of adolescents growing up in the Dutch area impacted by earthquakes induced by gas extraction". The topic is quite interesting, and the survey among adolescents, although the very reduced number of participants, seems well conducted. Notwithstanding this, I have some main concerns regarding presentation of the ms, mostly because of several sentences that are obscure to me.
We would like to thank the reviewer for their compliments and valuable comments. With the help of the comments, we think we have improved the manuscript.
- Although I am not a native English speking, what I fell in most paragraphs is that many redundant expressions are used, which make reading quite difficult. I would have appreciated the ms much more if, in general, the authors hade made a better effort toward schematizationof results. I can suggest to add a table at the beginning of the Discussion section to summarise the possible implications deduced from the answares to the different questions of the survey.
We thank the reviewer for these valuable suggestions. We have edited and shortened the text and included a conceptual model of our findings at the end of the results section.
- In as much, the authors left in the ms some paragraphs that do not pertain to it, like what I wonder wether is a part of Journal's Istruction to authors or not, or a couple of sentences that had already been written elsewhere in the text. Another problem I encountered concerns the drawings, which are recalled along the text, but that I did not find anywhere...possibly, it was my fault. The only attached material I found was a document in Dutch I cannot understand. If the drawings the authors make reference to are part of that document, I feel too difficult to find the right one in the present form.
Our apologies for this oversight on our part; we had accidentally left in the journal instructions. We have now deleted this paragraph. We had to submit the figures separately and therefore we only refer to the drawings in the manuscript. To clarify, we have now added the drawings in the revised version of the manuscript. We understand that you also received the Dutch-language report of the whole research project, which the editors asked us to send with the manuscript as background; this did not need to be reviewed. We apologize for any confusion.
Round 2
Reviewer 2 Report
Thank you for submitting the revised version. The authors have substantially improved their paper and now in my opinion it is almost ready to be published.
I recognized some minor issues to be corrected before giving the paper a go:
- You use italics fonts not consistently – please have a look at the page 11 – some sentences are partially in italics (even words, see line 489)
- Please consider adding reference to natural hazards which you mention in line 43-45 (i.e. for extreme weather events: doi 10.3390/su14042052)
- You have included participant drawings, which I recognize a great idea. Nevertheless, they are not visible enough, maybe it is an issue of poor scan quality. If it is easy to improve, please do it.
- Although you have strongly improved the text, please reread it once again to catch some minor spelling mistakes (i.e. line 273 – before word “finish” there is something left, an additional letter or a sign.
Author Response
- You use italics fonts not consistently – please have a look at the page 11 – some sentences are partially in italics (even words, see line 489)
Thank you for your critical eye. We rechecked the use of italics. We used it for titles of subsections, titles for tables and figures, and when we quote the participants of our study.
- Please consider adding reference to natural hazards which you mention in line 43-45 (i.e. for extreme weather events: doi 10.3390/su14042052)
We added this reference.
- You have included participant drawings, which I recognize a great idea. Nevertheless, they are not visible enough, maybe it is an issue of poor scan quality. If it is easy to improve, please do it.
It is quite difficult to improve the quality of the drawings but I think we succeeded. I think the problem with visibility refers to figure 1 and if the reviewer finds this version not visible enough, I think it is better to choose a more clear drawing of another participant.
- Although you have strongly improved the text, please reread it once again to catch some minor spelling mistakes (i.e. line 273 – before word “finish” there is something left, an additional letter or a sign.
We reread the manuscript and solved spelling mistakes.